# Molecular Characterization and SNP-Based Molecular Marker Development of Two Novel High Molecular Weight Glutenin Genes from *Triticum spelta* L.

**DOI:** 10.3390/ijms231911104

**Published:** 2022-09-21

**Authors:** Yuemei Cao, Junwei Zhang, Ruomei Wang, Haocheng Sun, Yueming Yan

**Affiliations:** Beijing Key Laboratory of Plant Gene Resources and Biotechnology for Carbon Reduction and Environmental Improvement, College of Life Science, Capital Normal University, Beijing 100048, China

**Keywords:** spelt wheat, HMW-GS, three-diamensional structure, SNP marker, quality improvement

## Abstract

Spelt wheat (*Triticum spelta* L., 2n=6x=42, AABBDD) is a valuable source of new gene resources for wheat genetic improvement. In the present study, two novel high molecular weight glutenin subunits (HMW-GS) 1Ax2.1* at *Glu-A1* and 1By19* at *Glu-B1* from German spelt wheat were identified. The encoding genes of both subunits were amplified and cloned by allele-specific PCR (AS-PCR), and the complete sequences of open reading frames (ORF) were obtained. *1Ax2.1** with 2478 bp and *1By19** with 2163 bp encoded 824 and 720 amino acid residues, respectively. Molecular characterization showed that both subunits had a longer repetitive region, and high percentage of α-helices at the N- and C-termini, which are beneficial for forming superior gluten macropolymers. Protein modelling by AlphaFold2 revealed similar three-diamensional (3D) structure features of 1Ax2.1* with two x-type superior quality subunits (1Ax1 and 1Ax2*) and 1By19* with four y-type superior quality subunits (1By16, 1By9, 1By8 and 1By18). Four cysteine residues in the three x-type subunits (1Ax2.1*, 1Ax1 and 1Ax2*) and the cysteine in intermediate repeat region of y-type subunits were not expected to participate in intramolecular disulfide bond formation, but these cysteines might form intermolecular disulfide bonds with other glutenins and gliadins to enhance gluten macropolymer formation. The SNP-based molecular markers for *1Ax2.1** and *1By19** genes were developed, which were verified in different F2 populations and recombination inbred lines (RILs) derived from crossing between spelt wheat and bread wheat cultivars. This study provides data on new glutenin genes and molecular markers for wheat quality improvement.

## 1. Introduction

Wheat *(Triticum aestivum* L., 2n=6x=42, AABBDD) is one of the three major food crops globally with a long history of cultivation, accounting for almost 35% of the human population’s dietary needs [1]. Wheat is an allohexaploid species, and has a huge genome (up to 17 GB) containing A, B, and D subgenomes and a large amount of repetitive sequences [2]. Wheat flour can be processed into various foods such as bread, noodles, cookies as well as non-food products because of its versatile functional features and excellent textural properties, which are closely related to the structure of gluten and interactions within the protein complex in grains [3,4].

The gluten proteins include water-insoluble monomeric gliadins and polymeric glutenins, which are considered as the largest protein molecule in nature [5]. Gliadins act as a “plasticizer” to endow dough extensibility or viscous flow. The glutenin proteins include high and low molecular weight glutenin subunits (HMW-GS, LMW-GS), of which HMW-GS serve as “backbone” for interactions with other glutenin subunits and confer dough elasticity or strength [6,7]. HMW-GS are encoded by two tightly linked paralogous genes (larger x- and smaller y-type) located at the *Glu-A1*, *Glu-B1* and *Glu-D1* loci on the long arm of the chromosome 1A, 1B and 1D, respectively. Theoretically, individual hexaploid wheat cultivars can express 6 different HMW-GS, but usually 3~5 HMW-GS are present due to gene silence [8,9]. The allelic variations at *Glu-1* loci are closely associated with flour processing quality [10]. For example, the subunit pairs and subunits 1Dx5+1Dy10, 1Bx7+1By8, 1Bx13+1By16, 1Bx17+1By18, 1Ax1, 1Ax2* have positive effects on gluten quality whereas 1Dx2+1Dy12 and 1Bx20 are related to poor dough strength [11,12,13].

Typical HMW-GS have three structural domains: a long and variable central repeat region conferring elasticity to protein molecules, and flanked by the short and conservative N- and C-terminals, in which most or all cysteine residues are present [14,15]. The repeat region is rich in β-turns flanked by spherical conservative regions formed by α-helices [16]. Most of the cysteine residues form intra-chain disulfide bonds, some of them form inter-chain disulfide bonds and directly affect dough formation and rheological characteristics [17,18,19]. Generally, the β-turn structure confers protein molecules with significant deformation resistance [20]. In particular, the central repeat domain can be folded in the action of protein disulfide isomerase, which is the basis of dough extension [21]. However, although different methods such as homology modelling and fold identification algorithm were used for deciphering the structure features of HMW-GS [22,23], the detailed 3D structures of wheat glutenin subunits and gluten proteins are still not clear due to the complexity of the protein compositions and the difficulty of crystallization [24,25,26]. The recently developed protein predicting tool Alphafold has provided the possibility to dissect the 3D structure of glutenin subunits, which is considered as the first computational method that can regularly predict protein structures with atomic accuracy even in cases in which no similar structure is known [27,28,29]. The recent study has shown that AlphaFold2 using the transformer design can obtain a high accuracy and is easy for 3D structure prediction of wheat MATE proteins [30].

In the past decades, a lot of studies have showed that the allelic variations at *Glu-1* in bread wheat are limited. However, more extensive *Glu-1* allelic variations are present in wheat related species such as spelt wheat [31,32], *T**. dicoccum* [33], *Ae. tauschii* [34,35], *Ae. longissima* [36,37], *Ae. speltoides* and *Ae. kotschyi* [38]. Spelt wheat (*T. spelta* L., 2n=6x=42, AABBDD) is closely related to common wheat and belongs to the same species with few differences among the subspecies [39,40,41]. As the first-class gene resource of wheat, spelt wheat is rich in nutritional value, various disease resistance factors, tolerance for various abiotic stresses and can be used for wheat genetic improvement through introducing the beneficial gene from spelt wheat [42,43,44,45]. In particular, some novel HMW-GS and subunit combinations in European spelt wheat varieties were identified such as 1Ax2.1*, 1Bx13*+1By19* and 1Bx6.1+1By22.1 [31]. These new allelic variations can be used as potential gene resources for improving wheat gluten quality. However, the molecular characterization of these novel glutenin genes and their application values for wheat quality improvement are still unknown.

To create novel germplasm for the improvement of wheat quality, we amplified and cloned the encoding genes of 1Ax2.1* and 1By19* subunits from European spelt wheat varieties by AS-PCR, and their molecular characterization and phylogentic relationships among HMW-GS genes were investigated. Meanwhile, AlphaFold modeling was used to reveal the 3D structural features of 1Ax2.1* and 1By19* subunits and previously characterized superior HMW subunits. On the basis of the sequence variations of promoter region, the specific SNP-based markers for *1Ax2.1** and *1By19** genes were developed and validated by using a wide range of wheat cultivars, F2 populations and RILs from different crossings between spelt wheat and bread wheat cultivars. The results are discussed in relation to provide new gene resources and molecular markers for the improvement of wheat’s breadmaking quality.

## 2. Results and Discussion

### 2.1. Identification of Novel HMW-GS in Spelt Wheat

HMW-GS compositions at *Glu-1* loci in Spelt 137 and Spelt 6 were identified by SDS-PAGE (Figure 1). The results showed that Spelt 137 contained 1Ax2.1*, 1Bx13+1By22* and 1Dx2+1Dy12 subunits at the *Glu-A1*, *Glu-B1* and *Glu-D1* locus, respectively. The new x-type subunit 1Ax2.1* encoded by *Glu-A1* located between 1Ax1 and 1Ax2* and had a lower electrophoretic mobility and higher molecular weight than 1Ax2* (Figure 1A). In addition, one new y-type subunit 1By22* was present at *Glu-B1* locus, which had a similar electrophoretic mobility and molecular weight with 1By18 subunit. The HMW-GS compositions in Spelt 6 were null at *Glu-A1*, 1Bx13*+1By19* at *Glu-B1* and 1Dx2+1Dy12 at *Glu-D1*. The new subunits 1Bx13* and 1By19* encoded by *Glu-B1* showed a similar electrophoretic mobility and molecular weight with 1Bx13 and 1By16 subunit, respectively (Figure 1B). The above two new subunits in spelt wheat were firstly found in our previous study [31]. These new allelic variations at *Glu-1* loci provide potential gene resources for wheat quality improvement.

### 2.2. Molecular Cloning and Characterization of 1Ax2.1* and 1By19* Genes from Spelt Wheat

Two specific amplification fragments with about 2500 bp from Spelt 137 and 2200 bp from Spelt 6 were obtained by AS-PCR (Appendix A), which were corresponding to the sizes of x-type and y-type HMW-GS genes, respectively. Both PCR fragments were collected, cloned and sequenced, and their complete coding sequences with typical structural characteristics of previously characterized HMW-GS genes were obtained and designated as *1Ax2.1** and *1By19**. Both *1Ax2.1** and *1By19** gene sequences were deposited in GenBank with the accession numbers of MK395158 and MK395159, respectively. The nucleotide sequences of *1Ax2.1** (2478 bp) and *1By19** (2163 bp) genes encoded 824 and 720 amino acid residues, respectively.

The deduced amino acid sequence of *1Ax2.1** gene showed four distinct domains as other x-type HMW-GSs in Figure 2: a signal peptide of 21 amino acid residues, an N-terminal sequence of 86 amino acid residues followed by a central repetitive domain of 672 residues, a C-terminal domain of 42 residues. The repeat region of 1Ax2.1* subunit consisted of tandem and interspersed repeats of 33 hexapeptide motifs (consensus PGQGQQ), 10 nonapeptide motifs (consensus GYYPTSPQQ and GYYPTSLQQ) and 23 tripeptide motifs (consensus GQQ). The deduced amino acid sequences of both genes were aligned with other HMW-GSs from wheat and related species to compare their sequence characteristics. Compared with the superior subunit 1Ax2* [46], 1Ax2.1* had 6 and 9 amino acid insertions, 6 amino acid deletions and 22 single amino acid substitutions, which ultimately increased the length of the repeat region. Differences in subunit size were mainly caused by the sequences of repetitive polypeptide motifs caused by base deletion or insertion [15,47]. It is known that the longer and more regular repeating units are usually more conducive to superior gluten quality [15,36]. 1Ax2.1* contained a longer repetitive domain with additional two hexapeptides (consensus PGQGQQ) than the superior quality subunit 1Ax2*. Besides, 1Ax2.1* had 6 amino acids deletion in repeat domain compared with the superior quality 1Ax1, but the tripeptides (GQQ), hexapeptides (PGQGQQ) and nonapeptides (GYYPTSPQQ) were highly consistent. And just like 1Ax2.1*, the number and distribution of cysteine residues of these subunits were highly conserved between them, including sites 31, 46, 61 at the N-terminal and site 812 or 818 or 803 at the C-terminal.

Similarly, *1By19** gene encoded 720 amino acid residues, and showed a similar primary structures with y-type HMW-GSs in Figure 3, including a signal peptide of 21 amino acid residues, an N-terminal of 104 amino acid residues, a repeat region of 553 amino acid residues and a C-terminal of 42 amino acid residues. The repeat region contained 8 nona- and 34 hexapeptides. In addition, 1Ax2.1* subunit contained four conserved cysteine residues (three at the N-terminus and one at the C-terminus) while 1By19* subunits had seven conserved cysteine residues (five at the N-terminus, one at the repetitive domain, and one at the C-terminus). Likewise, the repeat unit in 1By19* accounted for about 50% of the total amino acids in the repeat region, in which the number of peptide segment PGQGQQ was equivalent to the y-type superior quality subunit 1Dy10. Overall, 1By19* had the same length of amino acid sequence as the superior quality subunits 1By8 and 1By18. But the amino acid length of 1By19* located between the superior quality subunits 1By9 and 1By16 due to the insertion and deletion of amino acids in the middle repeat region. Besides, these subunits all contained the same number of nonapeptides (consensus GYYPTSLQQ), but 1By19* contained 1~2 additional hexapeptides (consensus PGQGQQ). And each of these subunits contained seven cysteine residues, including site 606 or 591 or 624 near the C-terminus flanked by sites 31, 43, 65, 66 and 76 at N-terminus and site 708 or 693 or 726 at the C-terminus.

### 2.3. SNP and InDel Variations in 1Ax2.1* and 1By19* Genes

Sequence variations resulting from point mutation and insertion/deletions (InDels) are the most common cause of wheat storage protein variation [48,49]. As listed in Table 1, 11 SNPs in *1Ax2.1** gene were present at different positions in N-terminal domain and in repeat region. Among them, 8 SNPs were nonsynonymous and resulted in the changes of corresponding amino acid residues of H-Y, L-S/P/G, E-G/T/Q, R-G, R-G/L, A-P/G, L-S/P. The remaining three SNPs at positions 198 bp (T-C), 288 bp (G-A) and 1302 bp (T-A/C) were synonymous and did not cause amino acid residue changes. One nonsynonymous SNP variation in *1By19** gene was detected: A/G-T transversion at the positions of 1137 bp, which led to the amino acid residue change (Q/E/G-H). No InDel variations were found in *1By19** gene.

### 2.4. Verification of the Cloned 1Ax2.1* and 1By19* Genes from Spelt Wheat by Tandem Mass Spectrometry Analysis

The corresponding HMW-GS on the SDS-PAGE gel were collected and digested, and then identified by MALDI-TOF/TOF-MS (Table 2). The results showed that 1Ax2.1* and the x-type subunit MK395158 could well match in five peptide segments with 35 (45–79), 24 (195–218), 49 (357–405), 6 (725–730) and 24 (792–815) amino acid residues, respectively (Figure 2). 1By19* was well marched to the y-type subunit MK395159 in one peptide segment of 43 residues (QLQCERELQESSLEACRQVVDQQLAGRLPWSTGLQMRCCQQLR) at the position of 28-70 amino acid (Figure 3). All peptide segments identified by tandem mass spectrometry acquired a high protein score (C.I.%) of 100, indicating a high realiability of MS/MS analysis. These results further verified the validity of the cloned *1Ax2.1** and *1By19** genes.

### 2.5. Phylogenetic Analysis of Ax2.1* and 1By19* Genes

The polymorphism of glutenin subunits is beneficial for understanding the origin and evolution of spelt wheat. The early study on the allelic variations at *Glu-1* and *Glu-3* in spelt wheat varieties supported the hypothesis of secondary origin of spelt wheat from hybridizing between cultivated emmer (*T. dicoccum*, AABB) and club wheat (*T. aestivum* subsp. *compactum*, AABBDD) [31]. Here, the complete encoding sequences of the cloned *1Ax2.1** and *1By19** genes from spelt wheat and other 27 HMW-GS genes from common wheat and related species were used to construct a neighbor-joining phylogenetic tree (Figure 4). The results showed that the x-type and y-type HMW-GS genes in the phylogenetic tree were clearly separated into two clades. In the x-type gene clade, 1Bx genes together with two Sx genes from S genome were classified into a subgrouop while the Dx and Ax genes were classified into another subgroup. In particular, *1Ax2.1** gene from spelt wheat showed a close phylogenetic relationship with *1Ax1*, *1Ax2** and *1Ax2.1* genes. In the y-type gene clade, the Dy and By genes were classified into different subgroups while *1By19** gene from spelt wheat was more closed to *1By16*, *1By9*, *1By8* and *1By18* genes.

To further explore the evolutionary relationships between *1Ax2.1** and *1By19** and other HMW-GS genes, their divergence times (million years ago, MYA) were estimated and the results are shown in Appendix A. In general, the divergent times between x-type subunit genes between y-type subunit genes at each locus were about 0-13 MYA and 0.2-6.0 MYA while the divergence between x- and y-type subunit genes in each locus occurred about 14-17 MYA. The divergence among *1Ax2.1** and *1Ax1* and *1Ax2** genes occurred more recently, at about 0.92-1.00 MYA while earlier divergence between *1Ax2.1** and *1Ax2.1* occurred at 2.31 MYA. *1By19** displayed a high similarity with *1By16* and they were diverged more recently at 0.31 MYA. It is known that the hexaploid wheat is originated from the hybridation between tetraploid wheat and *T. tauschii* that occurred at about ten-thousand years ago. Thus, *1Ax2.1** and *1By19** genes should emerge before spelt wheat formation.

### 2.6. Secondary Structure and 3D Structure Analysis of 1Ax2.1* and 1By19* Protein Subunits

The secondary structure characteristics of glutenin proteins can be used to uncover the molecular mechanisms of gluten quality formation [50]. In general, higher content of β-sheets and α-helix in glutenin subunits is helpful for forming good gluten structure and superior breadmaking quality [51,52]. In this study, the secondary structures of 1Ax2.1* and 1By19* subunits and other eight wheat HMW-GS (1Ax2*, 1Ax1, 1By8, 1By9, 1By15, 1By16, 1By18 and 1By20) were predicted by the PSIPRED server and a comparative analysis for their secondary structure features was performed. Among them, 1Ax2*, 1Ax1, 1By8, 1By15, 1By16 and 1By18 are considered as superior quality subunits whereas 1By20 is a poor quality subunit [11,12,13]. As shown in Table 3, 1Ax2.1* subunit had seven α-helixes (9.58%) and four β-strands (0.82%), better than the superior quality subunits 1Ax2* with the same seven α-helixes (9.33%) and two β-strands (0.98%), and 1Ax1 with six α-helixes (9.40%) and without β-strands. 1By19* subunit and other six y-type subunits generally contained eight α-helixes with different percentages, but no β-strands were present. 1By19* had 12.36% α-helixes, higher than 1By15 (12.03%), 1By16 (11.65%), 1By18 (12.22%), and 1By20 (11.85%), and lower than 1By8 (16.81%) and 1By9 (13.19%).

The 3D structures of 1Ax2.1* and 1By19* subunits and other six superior quality subunits 1Ax2*, 1Ax1, 1By8, 1By9, 1By16 and 1By18 were further predicted by AlphaFold2 (Figure 5). As a deep learning algorithm, AlphaFold2 can visualize, analyze and interpret 3D protein structures basing on amino acid sequences [53]. The results showed that Alphafold2 predictions with five cycles for eight HMW-GS were almost identical, including three indicators of alignment errors, per-residue confidence scores, matching templates (Appendix A). Eight subunits had different confidence scores in 3D structure models after five cycles of prediction, which were divided into five different ranks according to their scores (Appendix A). The highest level for eight subunits is shown in Figure 5, and the confidence score for 1Ax2.1* was 33.9, slightly close to 1Ax1 (34.9) and 1Ax2* (34.4). The confidence score for 1By19* was 36.9, slightly close to 1By16 (37.4), 1By9 (37.7), 1By8 (37.2) and 1By18 (37.0). All subunits predicted had a typical 3D structure, including two non-repetitive structural domains (N- terminal and C-terminal) with a repetitive fragment in the middle and a large number of α-helices at the N- and C-termini. In general, three x-type subunits (1Ax2.1*, 1Ax1 and 1Ax2*), and five y-type subunits (1By19*, 1By16, 1By9, 1By18 and 1By8) showed a similar 3D structural characteristics, respectively. It is worth noting that the number and distribution of Cys residues play an important role in determining the formation of glutenin polymer and the subsequent rheological parameters of dough [54,55,56]. Particularly, three x-type subunits (1Ax2.1*, 1Ax1 and 1Ax2*) contained four cysteine residues (three in the N-terminal domain and one in the C-terminal domain), which did not form intramolecular disulfide bonds within the subunits. Five y-type subunits (1By19*, 1By16, 1By18, 1By8 and 1By9) contained seven cysteine residues (five in the N-terminal domain, one in the C-terminal domain and one in the intermediate repeating structural domain), and formed three disulfide bonds: Cys 31~76, 43~65 and 66~706 in 1By19*, Cys 31~76, 43~65 and 66~726 in 1By16, Cys 31~76, 43-65 and 66~693 in 1By9, Cys 31~76, 43~65 and 66~708 in 1By8, and Cys 31~76, 43~65 and 66~708 in 1By8. However, the cysteines at the intermediate repeat region were unable to form a disulfide bond (Figure 5). Meanwhile, 1Ax2.1* and 1By19* subunits contained a large number of α-helices at the N- and C-termini, consistent with the above mentioned secondary structure prediction results as well as previous report [57,58]. Although the cysteines in 1Ax2.1*, 1Ax1 and 1Ax2* subunits and three cysteines in 1By19*, 1By16, 1By9, 1By8 and 1By18 subunits did not form intramolecular disulfide bonds, they might form intermolecular disulfide bonds with other glutenins or gliadins to further form gluten macropolymers [59].

### 2.7. Development and Validation of SNP-Based Molecular Markers for 1Ax2.1 * and 1By19* Genes

A new marker type named SNP, is only a bi-allelic type of marker in nature with low expectation of heterozygosity [60,61]. In the past years, a number of SNPs-based molecular markers for glutenin genes have been developed such as *1Dx2* and *1Dx5* alleles [62], *1By18* [49], *1Slx2.3** and *1Sly16** [36]. These markers can be for rapid improvement of wheat gluten quality. Herein, we amplified and cloned the upstream promoter sequences of *1Ax2.1** and *1By19** genes. Then BioEdit7.0 was used to compare their promoter sequence differences with other x- and y-type HMW-GS genes and the specific SNP variations were identified. Ultimately, we used these SNP variations to develop SNP-based molecular markers for *1Ax2.1** and *1By19** genes.

The specific primers were designed (Appendix A), and then used to amplify the upstream promoter sequences of *1Ax2.1** and *1By19** genes, and two specific fragments 905 bp and 728 bp from *1Ax2.1** and *1By19** genes were obtained, respectively (Appendix A). After cloning and sequencing, the typical promoter sequences of *1Ax2.1** and *1By19** genes were obtained, which were used for sequence alignment with the promoter sequences of other 12 x- and y-type HMW-GS genes deposited in GenBank (Appendix A and Appendix A). Two specific SNP sites in the upstream promoter region of each gene were identified: −377 bp and −203 bp in *1Ax2.1** and −591 bp and −56 bp in *1By19**. According to these specific SNP sites, two pairs of specific primers (2.1*F/R and 19*F/R) were designed (Appendix A) and used to develop SNP-based molecular markers of *1Ax2.1** and *1By19** genes as showed in Appendix A via combining with SDS-PAGE identification. The results from 47 bread wheat and spelt wheat varieties with different *Glu-1* allelic variations (Appendix A) showed that one 209 bp specific fragment was amplified by using 2.1*F/R primer pair from the varieties containing 1Ax2.1* subunit whereas no amplified products were obtained in the varieties without 1Ax2.1* subunit (Figure 6A,B). Similarly, one 570 bp specific fragment was amplified by the 19*F/R primer pair from varieties with 1By19* subunit, which was absent in the varieties without 1By19* subunit (Figure 6C,D). These results were well consistent with the SDS-PAGE identification. Further collection and sequencing of both amplified fragments also showed a consistence with their amplified regions as shown in Appendix A.

To further validate the developed SNP markers, two pairs of specific primers were used to amplify the promoter sequences of the target genes in different hybrid populations and RILs. Meanwhile, SDS-PAGE identification was conducted to verify the PCR results. For the marker validation, 200–250 grains from F2 generation populations and 50–80 grains from each RIL were detected. In the F2 populations of Spelt 137×Zhongmai 175, Spelt 137×Ningchun 4 and Spelt 137×Zhongmai 8601 crosses, 2.1*F/R primer could specifically amplify the 209 bp fragment in the F2 grains containing 1Ax2.1* subunit, and no amplified products were produced in the F2 grains without 1Ax2.1* subunit. At the same time, three F2 populations of Spelt 6×Zhongmai 175, Spelt 6×Ningchun 4 and Spelt 6×Zhongmai 8601 crosses were used to verify the molecular marker of *1By19** gene and the similar results were obtained. The F2 grains having 1By19* subunit showed the 570 bp specific fragment when amplified by 19*F/R primer pair (Figure 7). Furthermore, seven RILs from Spelt 137×Ningchun 4 crossing and six RILs from Spelt 6×Ningchun 4 crossing were further used to verify the developed molecular markers for *1Ax2.1** and *1By19** genes. As expected, all RILs containing 1Ax2.1* and 1By19* subunits could exhibit 209 bp and 570 bp specific fragments, respectively (Figure 8). These results were well consistent with SDS-PAGE identification, confirming that the developed SNP-based molecular markers have high specificity and accuracy for identifying *1Ax2.1** and *1By19** genes. Therefore, these SNP-based markers are expected to be used for breadmaking quality improvement via marker-assisted selection during wheat breeding program.

## 3. Materials and Methods

### 3.1. Plant Materials

The materials used in this work included two European spelt wheat varieties Spelt 6 and Spelt 137, 45 bread wheat and spelt wheat varieties with different allelic variations at *Glu-1* loci (Appendix A), six F2 generation populations from Spelt 137×Zhongmai 175, Spelt 137×Ningchun 4, Spelt 137×Zhongmai 8601, Spelt 6×Zhongmai 175, Spelt 6×Ningchun 4, and Spelt 6×Zhongmai 8601, seven and six F6 RILs, respectively, derived from crossing between Spelt 137×Ningchun 4 and Spelt 6×Ningchun 4 via consecutive self-crossing combining with screening and identification. All spelt wheat varieties were collected from Plant Breeding Institute, Technical University of Munich, Germany.

### 3.2. HMW-GS Extraction and SDS-PAGE

The extraction of seed HMW-GS was carried out with referred to the previous study with minor modifications [63]. Firstly, 70% ethanol (*v/v*) and 55% isopropanol (*v/v*) were sequentially added to homogenized seeds to remove albumins, globulins and gliadins. Then, glutenins were extracted by a commonly used glutenin extraction buffer (50% isopropanol, 80 μL Tris-HCl, pH 8.0) with 1% dithiothreitol (*w/v*) and 1.4% 4-vinylpyridine (*w/v*). The final extracted glutenin proteins were used for SDS-PAGE based on the reported method [34].

### 3.3. DNA Isolation, AS-PCR Amplication and Sequencing

The wheat seedling leaves were used to extract genomic DNA according to the improved CTAB method [37]. Two pair of AS-PCR primers were designed according to the published HMW-GS gene sequences and listed in Appendix A. The high-fidelity polymerases (Vazyme Biotech, Nanjing, China) were used for amplifying the complete coding sequence of HMW-GS. PCR reaction was performed by CFX96 Real Time system (Bio-Rad Laboratories) programmed at an initial denaturation at 95 °C for 3 min followed by 35 cycles of 95 °C for 15 s, 61 °C for 15 s, 72 °C for 150 s and finally extended at 72 °C for 5 min. The PCR products were separated by 1% agrose gel in Tris-acetic acid-EDTA buffer and the fragments of expected size were collected and purified by using Gel Extraction Kit (Omega, Bienne, Switzerland), and then the purified products were ligated into pMD18-T vector (TaKaRa Biotechnology, Dalian, China) and transformed into Trans-2-blue competent cells. Three positive clones were randomly selected to reduce the sequencing error, and then sequenced by TaKaRa Biotechnology, Dalian, China.

### 3.4. Sequence Alignment and SNP/InDel Identification

Multiple sequences alignments of the cloned HMW-GS genes and previously characterized x-type and y-type subunit genes were conducted by using Bioedit (version 7.0, Tom Hall, Scotts Valley, USA) SNPs and InDels variations in HMW-GS genes were identified. 15 x-type and 12 y-type HMW-GS genes deposited in GenBank are as follows: *1Ax1* (X61009), *1Ax2** (M22208), *1Ax2.1* (HQ834308), *1Bx6* (KX454509), *1Bx7* (X13927), *1Bx13* (JN982368), *1Bx14* (KF733216), *1Bx17* (AB263219), *1Bx20* (AJ437000), *1Bx23* (AY553933), *1Dx2* (KF466259), *1Dx2.2* (AY159367), *1Dx5* (KJ144185), *Sx1** (HQ380225), and *Sx3** (HQ380224), *1Ay* (FJ404595), *1By8* (AY245797), *1By9* (X61026), *1By15* (KF733215), *1By15** (KJ579440), *1By16* (EF540765), *1By18* (KF430649), *1By20* (LN828972), *1Dy10* (X12929), *1Dy12* (BK006459), *Sy9** (HQ380223), and *Sy18** (HQ380222).

### 3.5. MALDI-TOF/TOF-MS

According to the previously described method [49,51], the corresponding HMW subunits bands on SDS-PAGE gel were excised, and then digested with trypsin. HMW-GS identification of tandem mass spectrometry was performed by using EASY-nLC 1000 (Thermo/Finnigan, San jose, CA, USA) equipped with orbitrap Q Exactive mass spectrometry (Thermo/Finnigan).

### 3.6. Construction of Phylogenetic Tree and Estimation of Divergence Time

The full-length homologous nucleotide sequences were aligned using the Clustal W program, and the alignment file was used to construct phylogenetic tree according to the complete coding regions of HMW-GS by using software MEGA (version 6.0, Koichiro Tamura at al., Auckland, New Zealand). An evolution rate of 6.5 × 10^−9^ substitution/site year was used to estimate the divergence times of HMW-GS genes based on the reported method [64].

### 3.7. Secondary Structure and 3D Structure Prediction of HMW-GS

Secondary structure prediction of the deduced amino acid sequences from the cloned HMW-GS genes were carried out by PSIPRED (http://bioinf.cs.ucl.ac.uk/psipred/psiform.html, accessed on 10 May 2022) [65,66]. The 3D structures of HMW-GS were predicted by AlphaFold2 [67,68], and then editing was carried out by Pymol software (version 1.7.4, Schrödinger, Warren Lyford DeLano, New York City, USA).

### 3.8. Development and Validation of SNP-Based Molecular Markers

According to the SNP variations in the upstream promoter sequences of the coding region in the cloned HMW-GS genes (Appendix A), two pairs of AS-PCR primers were designed (Appendix A). Genomic DNA was extracted using the Trans Fast Taq DNA Polymerase system (TransGen Biotech, Beijing, China). PCR cycles consisted of 94 °C for 3 min for activation, followed by 35 cycles of 94 °C for 5 s, 58 °C/55 °C (Appendix A) for 15 s, 72 °C for 2/4 s, and finally ended at 72 °C for a 5 min extension step. The materials used for marker development and verification were described in the section of plant materials.

## 4. Conclusions

Two novel HMW-GS 1Ax2.1* and 1By19* were identified in European spelt wheat, and their complete encoding gene sequences of corresponding genes were cloned and sequenced. Molecular characterization showed that both subunits had longer and more regular repeating units, and a high percentage of α-helices and β-sheets. AlphaFold2 prediction showed that 1Ax2.1* and 1By19* showed similar 3D structural characteristics with three superior quality x-type subunits (1Ax2.1*, 1Ax1 and 1Ax2*) and four superior quality y-type subunits (1By16, 1By9, 1By8 and 1By18), respectively. Both subunits contained a large number of α-helices at the N- and C-termini. In particular, four cysteine residues in three x-type subunits (1Ax2.1*, 1Ax1 and 1Ax2*) and the cysteines in the intermediate repeat region were unable to form intramolecular disulfide bonds, but these cysteines might form intermolecular disulfide bonds with other glutenins and gliadins to enhance gluten macropolymer formation. The SNP-based molecular markers for *1Ax2.1** and *1By19** genes were developed according to the SNP variations in the promoter regions and verified in different F2 generation populations as well as different recombination inbred lines derived from crossing between spelt wheat and bread wheat cultivars. These molecular markers have shown a high reliability and have a good application prospect for improving bread making through marker-assisted selection.

## Figures and Tables

**Figure 1 ijms-23-11104-f001:**
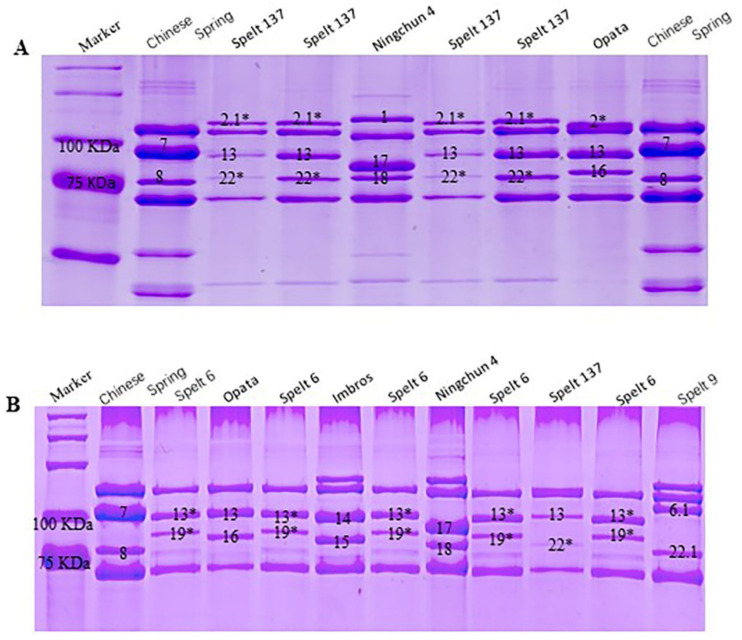
Identification of HMW-GS in European spelt wheat varieties Spelt 137 (**A**) and Spelt 6 (**B**) by SDS-PAGE. The HMW-GS in wheat is marked on the gel image with numbers or numbers with asterisks.

**Figure 2 ijms-23-11104-f002:**
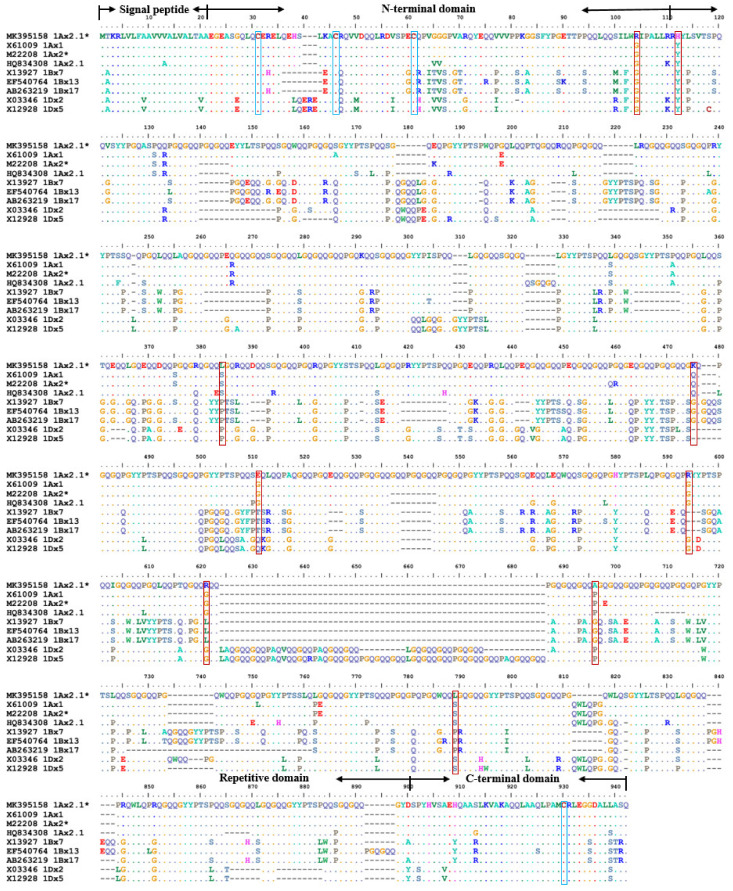
Comparison of the deduced amino acid sequences of *1Ax2.1** gene from spelt wheat with other eight x-subunit genes. The blue boxes indicate the cysteine residues, red boxes indicate differences of amino acid.

**Figure 3 ijms-23-11104-f003:**
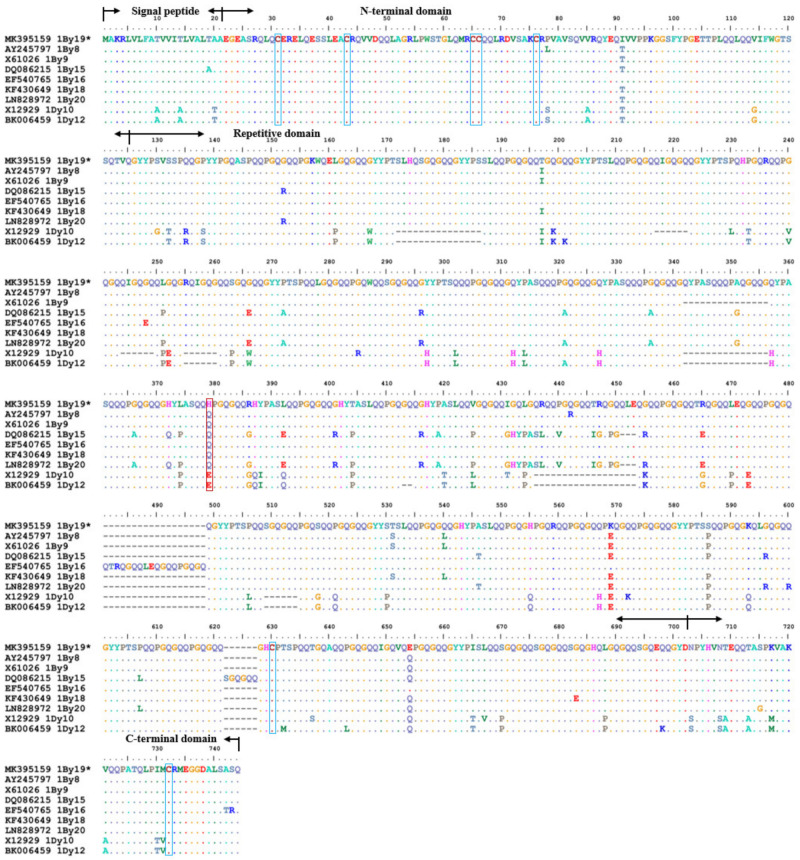
Comparison of the deduced amino acid sequences of *1By19** gene from spelt wheat with other eight y-type subunit genes. The blue boxes indicate the cysteine residues, red boxes indicate differences of amino acid.

**Figure 4 ijms-23-11104-f004:**
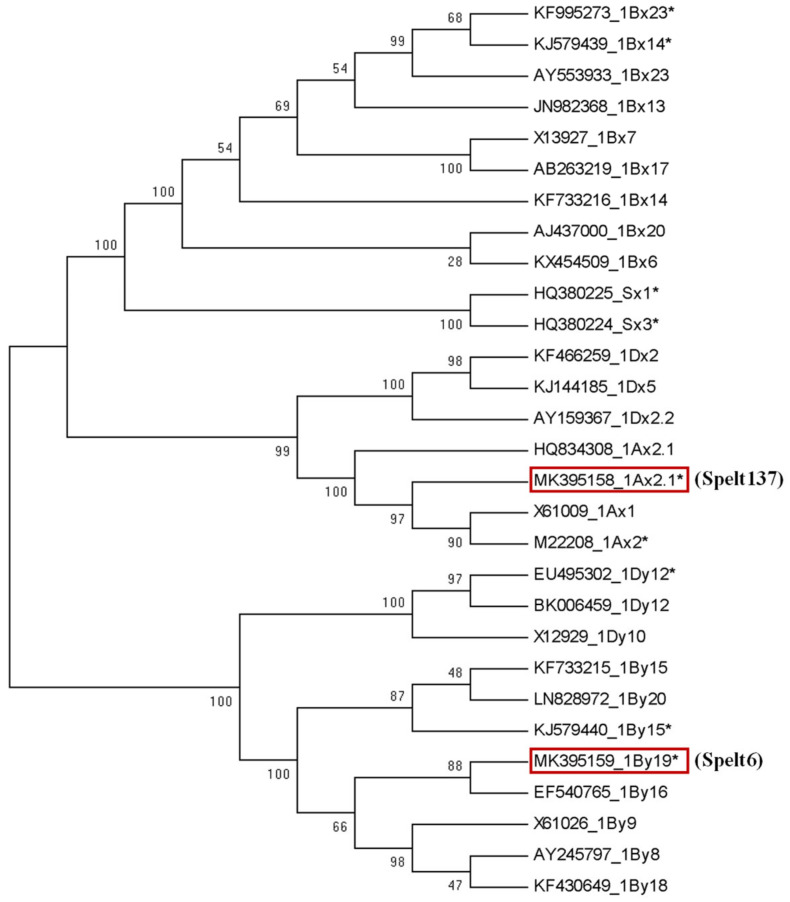
Phylogenetics of *1Ax2.1** and *1By19** genes and other 27 HMW-GS genes from A, B and D genomes of common wheat and related species. Bootstrap value ≥28% is indicated above or below the branches. The red frame indicates *1Ax2.1** and *1By19** genes, respectively. The 27 HMW-GS genes include X61009, M22208, KX454509, X13927, JN982368, KF733216, KJ579439, AB263219, AY553933, KF466259, AY159367, KJ144185, X61026, KF733215, KJ579440, EF540765, KF430649, X12929, BK006459 from *T*. *aestivum*; HQ380225 and HQ380224 from *Ae. speltoides*; KF995273 from *Triticum turgidum*; EU495302 from *T. aestivum* subsp. *Yunnanense*; AJ437000, LN828972 and AY245797 from *T. turgidum* subsp. *Durum*; HQ834308 from *T. monococcum* subsp. *Monococcum*.

**Figure 5 ijms-23-11104-f005:**
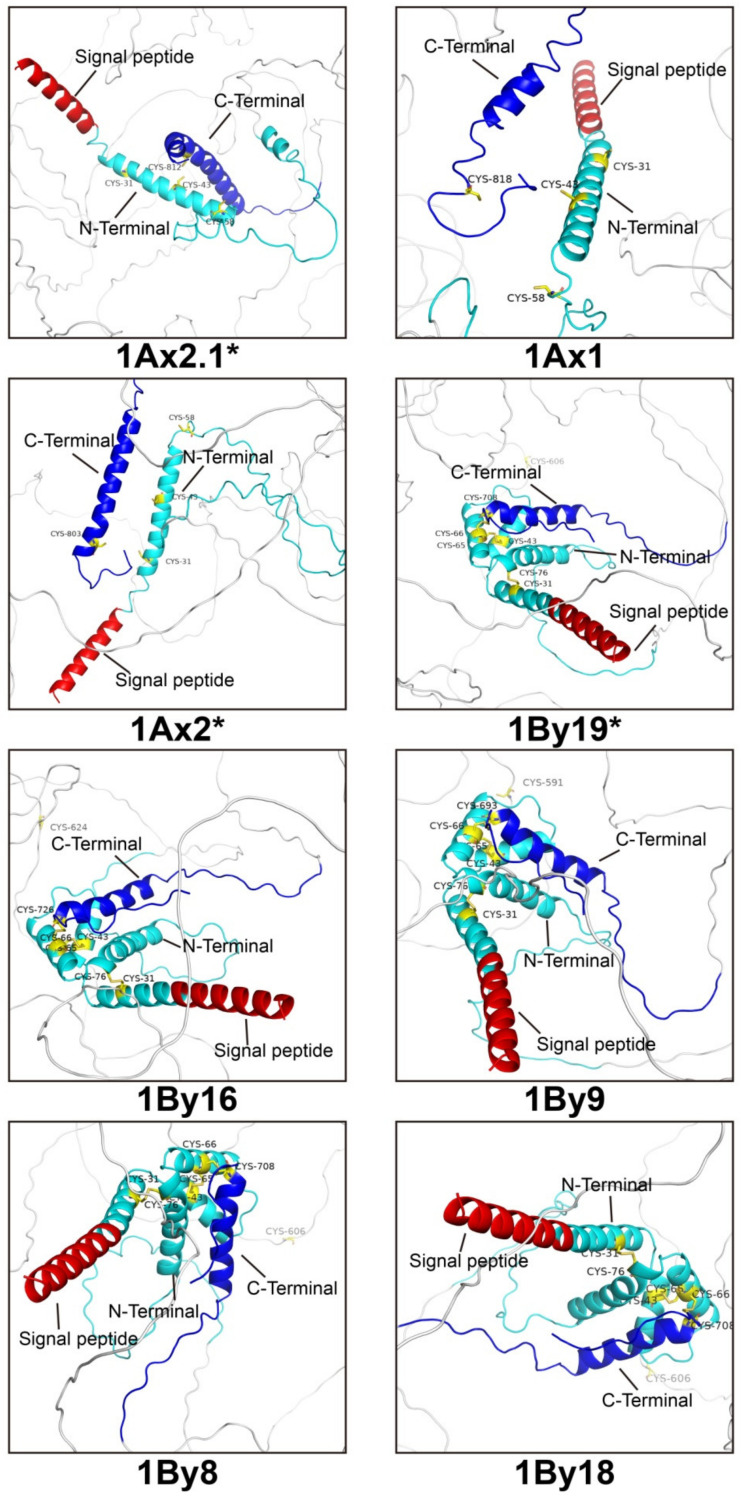
Localized view of the 3D structures of eight wheat HMW-GS by AlphaFold2.

**Figure 6 ijms-23-11104-f006:**
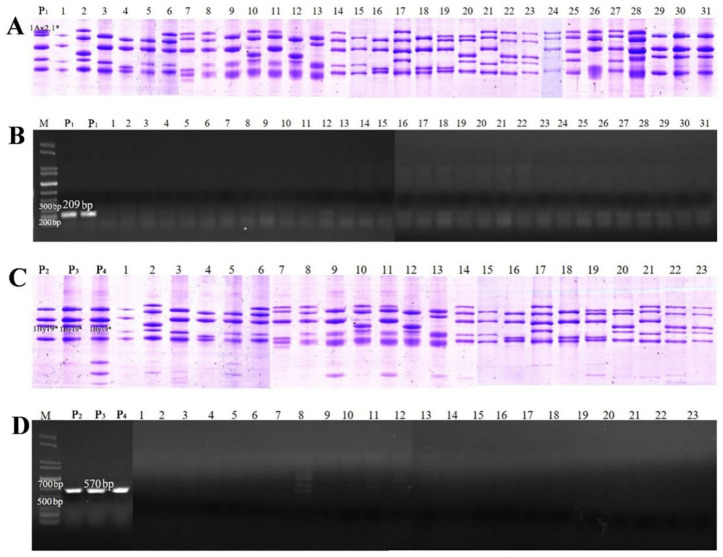
Development of the SNP-based molecular markers for *1Ax2.1** and *1By19** genes. (**A**) Identification of 1Ax2.1* subunit in different wheat varieties by SDS-PAGE. (**B**) PCR amplification of *1Ax2.1** gene from different wheat varieties by the primer 2.1*F/R. The 209 bp specific amplified fragment is indicated. (**C**) Identification of 1By19* subunit in different wheat varieties by SDS-PAGE. (**D**) PCR amplification of *1By19** gene from different wheat varieties by the primer 19*F/R. The 570 bp specific amplified fragment is indicated. The varied numbers in the figure are the same as those in Appendix A. P_1_: Spelt 137; P_2_: Spelt 6; P_3_: Spelt 20; P_4_: Spelt 24.

**Figure 7 ijms-23-11104-f007:**
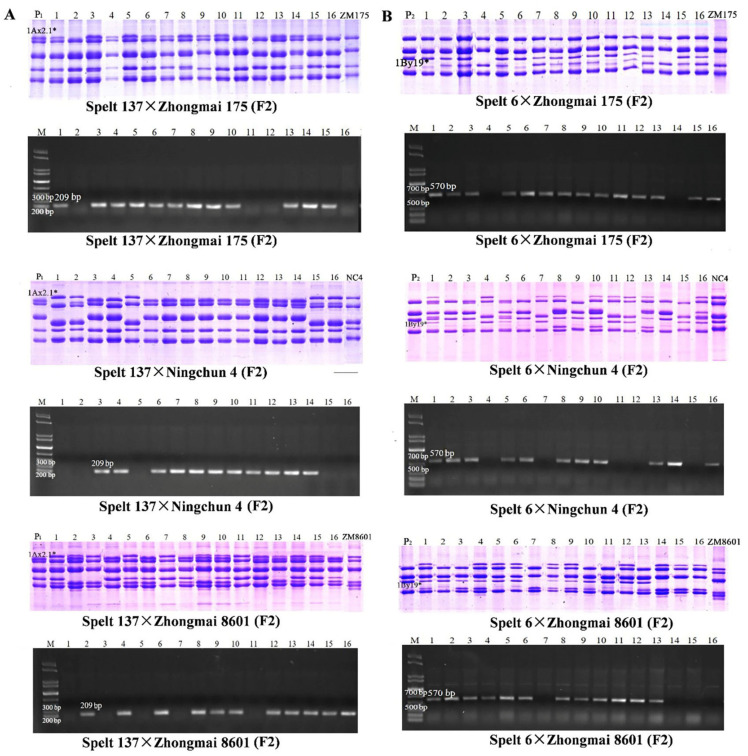
Validation of the SNP-based molecular marker for *1Ax2.1** (**A**) and *1By19** (**B**) genes in the F2 populations from crossing between spelt wheat and bread wheat varieties. 1Ax2.1* and 1By19* subunits and their special amplification fragments 209 bp and 570 bp are indicated. P_1_: Spelt 137; P_2_: Spelt 6; ZM175: Zhongmai 175; NC4: Ningchun 4; ZM8601: Zhongmai 8601.

**Figure 8 ijms-23-11104-f008:**
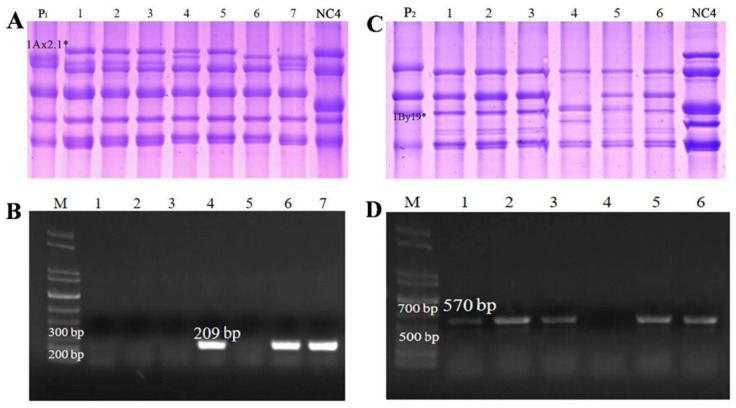
Validation of the SNP-based molecular markers for *1Ax2.1** and *1By19** genes in the RILs derived from Spelt 137×Ningchun 4 and Spelt 6×Ningchun 4 crosses. (**A**) HMW-GS from seven RILs (Lanes 1–7) from Spelt 137×Ningchun 4 identified by SDS-PAGE; (**B**) PCR amplification of *1Ax2.1** gene from seven RILs (Lanes 1–7) of Spelt 137×Ningchun 4; (**C**) HMW-GS from six RILs (lanes 1–6) from Spelt 6×Ningchun 4 identified by SDS-PAGE; (**D**) PCR amplification of *1By19** gene from six RILs (lanes 1–6) from Spelt 6×Ningchun 4. 1Ax2.1* and 1By19* subunits and the special amplification fragments 209 bp and 570 bp are indicated. P_1_: Spelt 137; P_2_: Spelt 6; NC4: Ningchun 4.

**Table 1 ijms-23-11104-t001:** Positions of SNPs identified between *1Ax2.1** and other x-type HMW-GS genes.

Positions (bp)	198	288	325	1076	1302	1448	1696	1777	1813	1815	2066
*1Ax2.1**	T	G	C	T	T	A	A	A	G	G	T
15 other x-type HMW-GS genes	C	A	T	C/--	A/C	G	G	G/--	C/T	A	C

**Table 2 ijms-23-11104-t002:** Identification of 1Ax2.1* and 1By19* subunits by MALDI-TOF/TOF-MS.

HMW-GS	Identified Protein	Accession No	Tryptic Fragments Identified by MS Data	Positions	Matched Peptides	Protein Score C. I. %
1Ax2.1*	x-type	MK395158	QVVDQQLRDVSPECQPVGGGPVARQYEQQVVVPPK QQPGQGQQLRQGQQGQQSGQGQPR QQDQQSGQGQQPGQRQPGYYSTSPQQLGQGQPRYYPTSPQQPGQEQQPR QWLQPR AQQLAAQLPAMCRLEGGDALLASQ	45–79 195–218 357–405 725–730 792–815	5	100
1By19*	y-type	MK395159	QLQCERELQESSLEACRQVVDQQLAGRLPWSTGLQMRCCQQLR	28–70	1	100

**Table 3 ijms-23-11104-t003:** Secondary structure prediction of 1Ax2.1* and 1By19* subunits and other eight wheat HMW-GS.

HMW-GS (Accession No.)	Type	Structure Motifs	Content (%)	Total	Amino Acid Length
1Ax2.1* (MK395158)	x-type	α-helix	9.58	7	824
		β-strand	0.82	4	
1Ax2* (M22208)	x-type	α-helix	9.33	7	815
		β-strand	0.98	2	
1Ax1 (X61009)	x-type	α-helix	9.40	6	830
		β-strand	0.00	0	
1By19* (MK395159)	y-type	α-helix	12.36	8	720
		β-strand	0.00	0	
1By8 (AY245797)	y-type	α-helix	16.81	8	720
		β-strand	0.00	0	
1By9 (X61026)	y-type	α-helix	13.19	8	705
		β-strand	0.00	0	
1By15 (DQ086215)	y-type	α-helix	12.03	6	723
		β-strand	0.00	0	
1By16 (EF540765)	y-type	α-helix	11.65	8	738
		β-strand	0.00	0	
1By18 (KF430649)	y-type	α-helix	12.22	8	720
		β-strand	0.00	0	
1By20 (LN828972)	y-type	α-helix	11.85	8	717
		β-strand	0.00	0	

## Data Availability

Not applicable.

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
