# Peer review of "Molecular Characterization and SNP-Based Molecular Marker Development of Two Novel High Molecular Weight Glutenin Genes from Triticum spelta L."

_ijms, 2022, doi:10.3390/ijms231911104_

Round 1
Reviewer 1 Report
This is original and interesting study about molecular characterization and SNP-based molecular marker development of two novel HMW glutenin genes from Triticum spelta L. The topic presents high interest because provides new gene resources and molecular markers for the improvement of wheat breadmaking quality.
The abstract abides by all the editing instructions and presents the objectives of the study. The introduction is well written and supported by well selected bibliographic data.
The manuscript presented for review is very well prepared. I have no substantive comments, I believe that the manuscript is refined and contains a number of results that achieve the purpose of the work as well as interesting, appropriate conclusions. My comments relate to some editorial oversight:
L. 31. Please write the scientific name in italics and check the entire manuscript.
L. 33. Write the word ″wheat″ with a capital letter after the period.
L. 96, 456. I suggest you write the expression ″breadmaking″ in two separate words.
L. 155-159. This sentence is too long and could not convey any meaning. In my opinion, this sentence should be restructured to bring out the meanings. If possible the statement can be broken down into smaller meaningful sentences.
L. 536. Please write the scientific name in italics.
Author Response
We respond to reviewers by presenting the document in Word.

Reviewer 2 Report
The reviewed paper is of high quality and represents the results of completely finished research having a clear practical value. Unfortunately, no supplements were available for review, only gel images were downloadable together with a manuscript file.
I recommend to provide some details to the figure/table captions (see file attached). In some places, data are better to be presented in tables (as they actually are) rather than in a form of list in text.
I also suggest to add more clear statement whether the designed SNP marker always cosegregated with the corresponding glutenin form or there were mismatches. As I may judge from Figure 6, there are protein bands on SDS-PAGE images which are not confirmed by the PCR screening. The discussed bands seem of very similar electrophoretic mobility with those of newly discovered glutenin forms.
Some comments and suggestions can be found directly in the file of manuscript (see attached). I hope these will be helpful for further improvement of a manuscript as well as for future work.
After making minor corrections, this manuscript can be accepted for publication in IJMS.

Author Response

(The authors gave the same response as above.)

Reviewer 3 Report
Identification and characterization of the subunits coding for high molecular weight glutenins is an important direction of research allowing for better use of wheat for nutritional purposes. in this work, the authors identified two new subunits and developed molecular markers for them. The work is valuable but requires a few corrections before being published.
Authors should clearly define the purpose of their research. in the last paragraph, introduction, they write what has been done at work - which looks more like conclusions than the purpose of the paper.
Why did the authors choose two spelled wheat? were they previously tested for glutenin content?
there is no description of the Maldi-TOF method in the materials and methods
Figure 1 is illegible, the names of the varieties should be listed in the legend, not in the photo
in chapter 2.2 the authors in the title write about the characteristics of genes and in the results they characterize the proteins encoded by these gans - it should be specified
Author Response

(The authors gave the same response as above.)
